# Ameliorative Effect and Mechanism of the Purified Anthraquinone-Glycoside Preparation from *Rheum Palmatum* L. on Type 2 Diabetes Mellitus

**DOI:** 10.3390/molecules24081454

**Published:** 2019-04-12

**Authors:** Fang-Rong Cheng, Hong-Xin Cui, Ji-Li Fang, Ke Yuan, Ying Guo

**Affiliations:** 1College of Pharmacy, Henan University of Chinese Medicine, Zhengzhou 450046, China; chengfr1963888@126.com (F.-R.C.); cuihongxin1974@163.com (H.-X.C.); 2Collaborative Innovation Center for Respiratory Disease Diagnosis and Treatment & Chinese Medicine Development of Henan Province, Zhengzhou 450046, China; 3Jiyang College of Zhejiang Agriculture and Forestry University, Zhu’ji 311800, China; fang_qiao418@163.com; 4Zhejiang Chinese Medical University, Zhejiang, Hangzhou 310053, China; littlegy@163.com

**Keywords:** *Rheum palmatum* L., type 2 diabetes mellitus, oxidative stress, apoptosis

## Abstract

*Rheum palmatum* L. is a traditional Chinese medicine with various pharmacological properties, including anti-inflammatory, antibacterial, and detoxification effects. In this study, the mechanism of the hypoglycemic effect of purified anthraquinone-Glycoside from *Rheum palmatum* L. (PAGR) in streptozotocin (STZ) and high-fat diet induced type 2 diabetes mellitus (T2DM) in rats was investigated. The rats were randomly divided into normal (NC), T2DM, metformin (Met), low, middle (Mid), and high (Hig) does of PAGR groups. After six weeks of continuous administration of PAGR, the serum indices and tissue protein expression were determined, and the pathological changes in liver, kidney, and pancreas tissues were observed. The results showed that compared with the type 2 diabetes mellitus group, the fasting blood glucose (FBG), total cholesterol (TC), and triglyceride (TG) levels in the serum of rats in the PAGR treatment groups were significantly decreased, while superoxide dismutase (SOD) and glutathione peroxidase (GSH-PX) levels were noticeably increased. The expression of Fas ligand (FasL), cytochrome C (Cyt-c), and caspase-3 in pancreatic tissue was obviously decreased, and the pathological damage to the liver, kidney, and pancreas was improved. These indicate that PAGR can reduce oxidative stress in rats with diabetes mellitus by improving blood lipid metabolism and enhancing their antioxidant capacity, thereby regulating the mitochondrial apoptotic pathway to inhibitβ-cell apoptosis and improve β-cell function. Furthermore, it can regulate Fas/FasL-mediated apoptosis signaling pathway to inhibit β-cell apoptosis, thereby lowering blood glucose levels and improving T2DM.

## 1. Introduction

Diabetes mellitus is an endocrine and metabolic disease characterized by hyperglycemia, often accompanied by a series of complications, including neuropathy, nephropathy, retinopathy, and cardiovascular and cerebrovascular disease [1,2]. It is a global disease affecting over 300 million people and is the fourth cause of death and disability in the world [3]. Notably, 90% of patients are type 2 diabetes mellitus (T2DM). The metabolic disorder of T2DM is associated with insulin resistance, which is linked to genetic, environmental interactions, and lifestyle [4]. Insulin resistance and insulin secretion defect are the two most recognized causes of T2DM [5]. Insulin resistance is caused by a decrease in the efficiency of insulin-induced glucose uptake and utilization, and insulin secretion defects are caused by insufficient insulin secretion due to apoptosis or dysfunction of β-cells of the pancreatic islets, resulting in the insulin which cannot meet normal physiological needs [6]. β-cell can regulate systemic metabolism by secreting insulin, the body’s only hormone lowering glucose [7]. Studies have shown that the mass of β-cell is lowered in the pancreas when there is an imbalance of regeneration and apoptosis [8]. There are many mechanisms inducing apoptosis of β-cells, including hyperglycemia, hyperlipidemia, oxidative stress, and activation of proinflammatory factors [9,10]. Therefore, protecting β-cells in the pancreas, inhibiting β-cell apoptosis, and increasing insulin secretion are important for T2DM patients [11].

*Rheum palmatum* L., a traditional Chinese medicine, has antibacterial, heat-clearing, and detoxifying properties, and has been used for the treatment of constipation and gastrointestinal diseases for more than 2000 years [12,13]. Although its chemical composition is relatively complex [14], it is known that anthraquinone, forming 3–5% of its content, is the most important activity component [15]. Modern pharmacological studies have shown that rhubarb-derived anthraquinones have a variety of physiological functions, such as antioxidant, antiviral, and anti-tumor properties, and protect cerebral cortex neurons [16,17,18]. In this paper, we studied the glucose-lowering effect of purified anthraquinone-Glycoside from *Rheum palmatum* L. (PAGR) in T2DM rats and explored the potential mechanism in the context of lipid metabolism, oxidative stress, and apoptosis.

## 2. Results

### 2.1. Anthraquinone-Glycosides Content of PAGR

The prepared working solution was analyzed three times under 4.3.1 chromatographic conditions of high-performance liquid chromatography (HPLC), and the contents of the three anthraquinone-Glycosides were determined (Figure 1). The concentrations of emodin-8-*O*-β-d-glucoside, aloe-emodin-8-*O*-β-d-glucoside, and chrysophanol-8-*O*-β-d-glucoside were calculated using the previously determined standard curve and determined to be 6.36%, 8.13%, and 6.78%, respectively, while the relative standard deviation (RSD) was 1.87%, 2.02%, and 1.58% respectively. The results showed that these three anthraquinone-Glycosides make up more than 20% of the total anthraquinone-Glycosides, and aloe-emodin-8-*O*-β-d-glucoside was the major anthraquinone-Glycoside among them.

### 2.2. Acute Toxicity Study

The acute toxicity of PAGR was evaluated at a dose of 2000 mg/kg. There were no deaths or obvious symptoms of poisoning within the initial 24 h of close observation and no behavioral abnormalities over the following 14 days. Therefore, the doses of PAGR for the low, middle (Mid), and high (Hig) groups were selected as 100, 200, and 400 mg/kg, respectively.

### 2.3. Changes inBody Weight and Fasting Blood Glucose in Rats

Polydipsia, polyuria, polyphagia, weight loss, and hyperglycemia are the most typical symptoms of T2DM [19]. The changes in rats’ body weight and fasting blood glucose (FBG) during the administration of streptozotocin (STZ) and PAGR are shown in Figure 2. The body weight of rats treated with STZ and a high-fat diet was obviously decreased, while FBG levels were markedly increased. After administration of metformin (Met) and PAGR, the body weight and FBG levels of rats began to recover. Notably, after six weeks of treatment, the body weight of the rats in the Met and PAGR treatment groups was significantly higher than that of the T2DM group (*p* < 0.05), and their FBG concentration was significantly lower than the T2DM group (*p* < 0.05). Furthermore, the Hig treatment group had returned to close to normal levels.

### 2.4. Effect of PAGR on Lipid Metabolites and Antioxidant Enzyme Activities in Rats

The effect of PAGR on triglyceride (TG) and total cholesterol (TC) is shown in Figure 3a,b. Compared with the normal (NC) group, the TC and TG of the T2DM group were significantly increased (*p* < 0.01). However, compared with the T2DM group, the TC and TG of the Met and PAGR treatment groups were significantly decreased in a dose-dependent manner. The specific dates were shown in Appendix A.

By measuring the concentration of superoxide dismutase (SOD) and glutathione peroxidase (GSH-PX) in the serum of rats, it was found that the SOD and GSH-PX levels in the T2DM group were significantly decreased when compared with the NC group (*p* < 0.05), while those in the Met and PAGR treatment groups were markedly increased when compared with the T2DM group, with significant differences between the Met and PAGR Hig groups (*p* < 0.05) (Figure 3c,d).

In the course of the development of T2DM, the renal tissue is damaged, and its functions were altered due to long-term of dysglycemia [20]. It is shown as Figure 3e,f that the serum creatinine and blood urea nitrogen (BUN) in the T2DM group were significantly increased compared with the NC group (*p* < 0.01), while the serum creatinine and BUN in the Met and PAGR treatment groups were significantly decreased when compared with T2DM group, indicating that PAGR has a protective effect on the kidney.

### 2.5. The Effect of PAGR on Histopathological Changes

#### 2.5.1. Effects of PAGR on Histopathological Changes in Liver

Hyperglycemia not only affects the transduction of apoptotic signals in islet cells but also damages tissues and organs [21]. As one of the major metabolic organs, the liver is the main organ damaged by diabetes. Steatosis, inflammatory cell infiltration, and necrosis are the main characteristics of hyperglycemia-induced liver injury [22]. As evident in Figure 4, the structure of liver tissue of rats in the NC group is intact, and the hepatocytes are arranged radially around the central vein. In contrast, compared with the NC group, the number of hepatocytes in the T2DM group is reduced, the hepatic cord is disordered, and hepatocytes show obvious degeneration and necrosis. The hepatocytes surrounding the central vein are filled with lipid droplets, and the infiltration of inflammatory cells is also evident. Notably, these histopathological changes in the livers of the Met and PAGR treatment groups were improved to varying degrees when compared with the T2DM group.

#### 2.5.2. Effects of PAGR on Histopathological Changes in Kidney

Reduction of SOD and GSH-PX levels and an increase of lipid peroxide and free radicals in vivo can all increase the expression of transforming growth factor (TGF)-β1 [23]. TGF-β1 can inhibit cell proliferation, promote renal cell hypertrophy, and lead to glomerulosclerosis and tubule interstitial fibrosis [24]. It can be seen in Figure 5 that the renal tissue structure was distinct and clear and the glomerular structure was normal in the NC group. In contrast, the glomerular volume of the model group was enlarged, and the basement membrane was thickened when compared with the NC group. However, compared with the T2DM group, the glomerular volume of the Met and PAGR treatment groups was smaller, the mesangial matrix had slight hyperplasia, and the histopathological changes were obviously improved.

#### 2.5.3. Effects of PAGR on Histopathological Changes in Pancreas

In addition to persistent hyperglycemia, patients with type 2 diabetes often have accompanying hyperlipidemia, and long-term exposure to high concentrations of glucose or lipids not only causes functional disorders of islet cells but also disrupts the islet structure [25]. It can be seen in Figure 6 that NC group rats showed normal histology. The islet shape was regular. Islet cells formed clumps and were distributed among the pancreas exocrine glands. Compared with the NC group, the islet structure in the T2DM group showed obvious deformation, fewer islet cells, blurred boundaries, and the structure was unclear. Compared with the T2DM group, the histopathological changes present in the islet tissue of the Met and PAGR treatment groups were attenuated to some degree.

### 2.6. Effect of PAGR on the Expression of Cytochrome C (Cyt-c), Caspase-3, and FasLin Pancreas

As shown in Figure 7, the expression of FasL, Cyt-c, and caspase-3 in pancreatic tissue of rats after administration of PAGR and metformin was significantly reduced compared with the T2DM group.

## 3. Discussions

STZ is the most commonly used hydrophilic compound for inducing T2DM. It can be transported into the cell membrane through the glutamine transaminase transporter to induce DNA alkylation. DNA alkylation can induce β-cell death [26], which consequently leads to hyperglycemia. Moreover, a high-fat diet plays an important role in insulin resistance [27]. The gradual natural progression and changes in the metabolism of human patients with T2DM are well mimicked by a high-fat diet and low-dose STZ in rats [28].

There is an inseparable relationship between glucose metabolism and lipid metabolism. Hyperglycemia can lead to dyslipidemia, while abnormal lipid metabolism is considered a major risk factor for diabetes and its multiple complications [29]. Lipid metabolites, such as TG and TC, directly antagonize insulin signaling and are considered the main cause of insulin resistance [30]. When TG remains high, heparin activates lipoprotein lipase, increasing intravascular lipolysis of TG, thereby increasing the exposure of tissues to free fatty acids, leading to insulin resistance and impairing β-cell function [31]. TC is the sum of cholesterol contained in all lipoproteins in the blood and is closely related to various diabetic complications, including cardio-cerebral vascular disease and neuropathy [32]. Therefore, improving lipid metabolism may ameliorate diabetes and its complications [33].

Hyperglycemia can also induce oxidative stress and lipid peroxidation. Importantly, oxidative stress can regulate insulin secretion in different ways and accelerate the development of diabetes mellitus [34]. For example, increased oxidative stress may have a negative effect on the regulation of blood glucose and cause dysfunction or apoptosis of glucose-regulating cells, such as β-cells, by stimulating the stress-responsive pathway for regulation [35]. Oxidative stress stimulates mitogen-activated protein kinase (MAPK) stress signals and causes inhibition of insulin signaling [36]. In addition, oxidative stress can promote the expression of many proinflammatory factors, including tumor necrosis factor (TNF)-α and interleukin (IL)-6, and significantly decrease insulin sensitivity [37]. Furthermore, studies have shown that there is a direct interaction between oxidative stress and insulin resistance, and the accumulation of oxidation products may damage critical macromolecules in insulin-sensitive tissues [38]. Therefore, from the perspective of treatment, reducing oxidative stress in the body may ameliorate diabetes. SOD and GSH-PX are two important enzymes in the antioxidant system, which can reflect the body′s antioxidant capacity. It was found in our study that PAGR could improve glucose and lipid metabolism in diabetic rats and reduce oxidative stress. Therefore, it may act by improving the function of insulin secreting β cells.

Under normal physiological conditions, the number of β-cells in the pancreas is in a dynamic equilibrium due to the regulation of apoptosis, proliferation of pancreatic islet, and production of new insulin by secretory tube. However, diabetes can develop when β-cell apoptosis occurs in excess [29]. There are two main pathways of apoptosis, the intrinsic (mitochondrial driven) pathway and the extrinsic (receptor-mediated) pathway [39]. Both oxidative stress and abnormal lipid metabolism promote the production of reactive oxygen species (ROS) and reactive nitrogen species (RNS). ROS and RNS can change the membrane potential of mitochondria, leading to the release of Cyt-c [40]. The release of Cyt-c activates caspase-3 [41], which is the final effector caspase in the caspase cascade and is the common downstream effector of multiple apoptotic pathways [42]. Caspase-3 induces apoptosis and thereby drives the death of β-cells resulting in insufficient insulin secretion. Correspondingly, the process of apoptosis is accompanied by a change in mitochondrial function and structure, which will also lead to the leakage of Cyt-c. As a marker of mitochondrial damage, Cyt-c levels can reflect the degree of mitochondrial structural damage [43]. Mitochondria act as energy transducers for cells, and their destruction can result in the abnormal function of islet β-cells, as shown in Figure 8. The expression of Cyt-c and caspase-3 corresponded with the increased levels of SOD and GSH-PX in the serum. PAGR can improve the antioxidant capacity of rats with diabetes and thus reduce the level of oxidative stress, regulating the mitochondrial-induced apoptosis pathway and reducing the damage to the mitochondrial structure, thereby protecting the pancreatic β-cells from apoptosis and restoring the function of β-cells.

High concentration of glucose also promotes the expression of Fas and FasL in pancreatic tissue [44]. FasL is an important ligand for inducing cell death and is a signaling factor of the extrinsic apoptosis pathway [45]. When it binds to Fas, which is the death receptor, it induces the assembly of a series of proteins that induce the death signaling complex in seconds. These proteins then activate procaspase-8, leading to activation of the caspase cascade, which finally induces the activation of caspase-3 [46]. When the caspase cascade is activated, it induces apoptosis swiftly [47], resulting in decreased islet β-cells, which is the basis for the decline of insulin secretion. The results of western blotting suggested that PAGR could inhibit apoptosis of β-cells and improve insulin secretion by regulating the Fas/FasL-mediated apoptotic signal pathway.

## 4. Material and Methods

### 4.1. Drugs and Chemicals

*Rheum palmatum* L. medicinal materials were purchased from Zhejiang University of Traditional Chinese Medicine Chinese Herbal Pieces Co., Ltd. (Zhejiang, China) and identified by the Professor Jiawei Huang who is from Zhejiang Chinese Medical University. STZ was purchased from Aladdin Bio-reagent (Shanghai, China). Glucose, Superoxide dismutase, Glutathione peroxidase, Triglyceride, Total cholesterol, Serum creatinine, Blood urea nitrogen Kits were purchased from Nanjing Jiancheng Biotechnology Co., LTD (Nanjing, China). The BCA Kit was purchased from Aidlab Biotechnologies Co., LTD (Beijing, China). The antibodies specific to FasL, Cyt-c, caspase-3 were purchased from Xinbosheng Technology Co., LTD (Shanghai, China). Emodin-8-*O*-β-d-glucoside, aloe-emodin-8-*O*-β-d-glucoside, chrysophanol-8-*O*-β-d-glucoside substances were prepared in the laboratory, the structure was identified by ^1^H NMR, ^13^C-NMR, and MS, and the purity was over 98% calibrated by HPLC peak area normalization method. All other reagents used were also chromatographically or analytically pure.

### 4.2. The Preparation and Determination of PAGR

Ten kg roots of *Rheum palmatum.* L. medicinal material was weighed and crushed and passed through a 60-mesh sieve. Then, it was extracted by reflux extraction three times for 1.5 h each, with a 1:7 ratio of plant material to 80% ethanol, followed by extraction filtration. The filtrate was combined and concentrated under reduced pressure at 60 °C bya rotary evaporator (RV3V, Staufen, Germany) until no alcohol could be smelled, and the appropriate amount of water was added for ultrasonic dissolution. A concentrate of 6000 mL was obtained. The concentrate was extracted three times with petroleum ether, ethyl acetate, and n-butanol at a volume ratio of 2 times, and the extraction liquid was combined. Then, it was concentrated under vacuum pressure to obtain the dry powder from each extracted fraction, in which the petroleum ether fraction was 158.4g, the ethyl acetate fraction was 212.7 g, the n-butanol fraction was 343.6 g, and the water was 425.9 g. The dry powder from the n-butanol fraction was dispersed in water by ultrasound and enriched and purified using a macroporous resin column (Diaion HP-20, Tokyo, Japan). First, it was eluted with distilled water, followed by 10%, 20%, 40%, and 60% methanol. Each eluted fraction was collected and concentrated under reduced pressure to dryness to obtain dry powder from each elution fraction. The total anthraquinone-Glycoside content in each elution fraction was determined by ultraviolet absorption spectrometry (UV-1801, Beijing, China) with emodin-8-*O*-β-d-glucoside as the standard substance, and the content was 4.63%, 8.84%, 21.91%, and 6.32%, respectively, determined by ultraviolet and visible spectrophotometer (UV-1801, Beijing, China). The anthraquinone-Glycoside was mainly concentrated in the 40% methanol elution fraction. Therefore, it was selected as the active fraction for the following experiments and is henceforth referred to as PAGR.

### 4.3. Determination of Three Anthraquinone-Glycosides in PAGR by HPLC

#### 4.3.1. HPLC Chromatographic Conditions

Agilent 1200 (Santa Clara, CA, USA) High-Performance Liquid Chromatography consists of a quaternary pump, autosampler, column oven, and VWD detector. Column: Agilent Extend-C18 (4.6 mm × 250 mm, 5 μm). The column temperature was 30 °C, flow rate 1.0 mL/min, detection wavelength 280 nm. Mobile phase acetonitrile was 0.1% formic acid (A) and acetonitrile (B), 0–10 min, A: 90%, B: 10%, 10–25 min, A: 88%, B: 12%, 25–30 min, A: 78%, B: 22%, 30–40 min, A: 67%, B: 33%, and the injection volume was 5 μL.

#### 4.3.2. Preparation of Working Solution and Standard Curve

A certain amount of PAGR was accurately weighed, dissolved by ultrasound, and a working solution was prepared in methanol at constant volume. Certain amounts of emodin-8-*O*-β-d-glucoside, aloe-emodin-8-*O*-β-d-glucoside, and chrysophanol-8-*O*-β-d-glucoside were dissolved in methanol to form a mixed reference solution, which was then diluted with methanol to prepare mixed standard solutions with concentrations of 0.5200, 0.2600, 0.1300, 0.0625, 0.0163, 0.0081, 0.0040, and 0.0020 mg/mL respectively. The peak area of the different concentrations of the mixed standard solution was determined by HPLC, and the standard curve was prepared with the peak area as the ordinate and the mixed standard solution concentration as the abscissa. It was determined that the regression equation of emodin-8-*O*-β-d-glucoside was y = 14932x + 1.496, R^2^ = 0.999, aloe-emodin-8-*O*-β-d-glucoside was y = 96120x + 1.555, R^2^ = 0.999, and chrysophanol-8-*O*-β-d-glucoside was y = 26804x − 1.231, R^2^ = 0.999, indicating that the three anthraquinone-Glycosides had a good linear relationship in the concentration range of 0.0020–0.5200 mg/mL.

#### 4.3.3. Stability and Precision

A certain concentration of the mixed reference solution was injected six times within a day according to the chromatographic conditions to determine the peak area. The RSD of the peak areas of emodin-8-*O*-β-d-glucoside, aloe-emodin-8-*O*-β-d-glucoside, and chrysophanol-8-*O*-β-d-glucoside was1.18%, 1.52%, and 2.06%, respectively, which indicated that the precision of the instrument was good. The reference solution was placed at room temperature and analyzed at 2, 4, 6, 8, and 10 h according to the chromatographic conditions, and the RSD of the peak area of emodin-8-*O*-β-d-glucoside, aloe-emodin-8-*O*-β-d-glucoside, and chrysophanol-8-*O*-β-d-glucoside was 2.30%, 1.26%, and 1.20%, respectively, indicating that the sample was stable within 10 h.

### 4.4. Acute Toxicity Test

A single dose of 2000 mg/kg of PAGR was intragastrically administered in rats. The death and poisoning symptoms of the rats were observed closely within the first 24 h after administration, and their behavior was monitored routinely over the next 14 days to evaluate the acute toxicity of PAGR.

### 4.5. AnimalsTreatment and Sample Collection

Male Sprague-Dawley (SD) rats (150 ± 10 g) were purchased from the Experimental Animal Center of Zhejiang Academy of Medical Sciences (SCXK, 2015-0033, Zhejiang, China). All experimental animals were housed in a standard animal room at a temperature of 23 °C ± 1 °C and humidity of 50–60%. Rats were adapted to the environment for one week before the experiment and were then randomly divided into six groups (*n* = 10): NC group, T2DM group, Met treated group, and Low, Mid, and Hig dose groups of PAGR treated. Except for the NC group, the rats in the other groups were fed with a high-fat diet (corn starch 60%, casein 20%, soybean oil 20%) [48] for 6 weeks and then injected intraperitoneally with STZ (30 mg/kg) two times every other day. FBG was measured after 72 h of the last injection and was >11.1 mol/L, indicating that the diabetes model was successful. Next, the NC and T2DM groups were intragastrically administered with normal saline, Met group was administered with 100 mg/kg metformin [49], and the Low, Mid, and Hig groups were treated with 100, 200, and 400 mg/kg of PAGR for 6 weeks. They all were injected once a day. Blood samples were drawn from the orbit to measure FBG every week. After fasting for 12 h after the last administration of PAGR, rats were anesthetized with 10% chloral hydrate, the blood was taken from the abdominal aorta, and the liver, kidneys, and pancreas were removed quickly and stored at -80°Cfor further study. All experimental procedures were in accordance with the Guide for the Care and Use of Laboratory Animals of Zhejiang and were approved by the Committee on the Ethics of Animal Experiments at Animal Center of Zhejiang Agriculture and Forestry University [Ethics Certificate No. 2014001812184].

### 4.6. Biochemical Measurements

The SOD, GSH-PX, TG, and TC of serum were measured by Superoxide dismutase, Glutathione peroxidase, Triglyceride, and Total cholesterol Kits of Nanjing Jiancheng, respectively, according to the commercial instruction.

### 4.7. Histopathological Examinations

The liver, pancreas, and kidneys were fixed in 10% formalin solution for 24 h and then paraffin-embedded. Paraffin sections of 5–7 μm thickness were carved up and stained with hematoxylin and eosin (H&E), and histopathological changes were observed under a BX20 optical microscope (Tokyo, Japan).

### 4.8. Western Blot Analysis

The pancreatic sample was lysed using RIPA lysis buffer on ice and centrifuged at 10,001× *g* (4 °C, 10min). The protein concentration of the supernatants was determined by the BCA Kit. For western blot analysis, the protein was separated by sodium dodecyl sulfate polypropylene gel electrophoresis (SDS-PAGE) and then electrotransferred to polyvinylidene difluoride (PVDF) membranes. All membranes were incubated with 50 g/L non-fat milk powder in triethanolamine buffer for 1 h at room temperature and then with primary antibodies against FasL, Cyt-c, caspase-3 (diluted 1:1000), and β-actin (diluted 1:1500) at 4 °C overnight. The following day membranes were washed with tris-buffered saline containing Tween-20 (TBST), then incubated with peroxidase-conjugated anti-rabbit secondary antibody for 1 h at room temperature. After washing with TBST three times, ECL reagent was added, and images were captured using a chemiluminescence detection system (Amersham Pharmacia, Piscataway, NJ, USA).

### 4.9. Statistical Analysis

All data were expressed as mean ± SD and analyzed by the SPSS statistical software (SPSS19.0 Inc., Chicago, IL, USA). One-way ANOVA with Duncan’s test was used for inter-group comparison. *p*-values < 0.05 were considered statistically significant, and *p*-values < 0.01 were considered extremely significant.

## 5. Conclusion

In conclusion, PAGR has glucose-lowering properties and attenuates type 2 diabetes. The potential mechanism is that PAGR reduces the level of oxidative stress by improving lipid metabolism and enhancing antioxidant capacity, which reduces damage to mitochondrial structures and downregulates activation of mitochondrial-induced cell death pathways; thereby inhibiting β-cell apoptosis and improving β-cell function. It also downregulates Fas/FasL-mediated apoptosis in pancreatic tissue, further inhibiting apoptosis of β-cells.

## Figures and Tables

**Figure 1 molecules-24-01454-f001:**
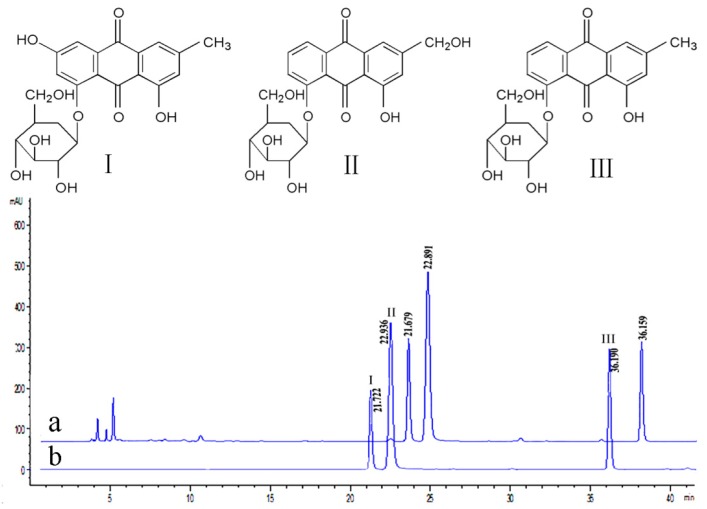
The structure of the emodin-8-*O*-β-d-glucoside (**I**), aloe-emodin-8-*O*-β-d-glucoside (**II**), and chrysophanol-8-*O*-β-d-glucoside (**III**) and HPLC chromatogram of purified anthraquinone-Glycoside from *Rheum palmatum* L. (PAGR) (**a**) and mixed Reference (**b**).

**Figure 2 molecules-24-01454-f002:**
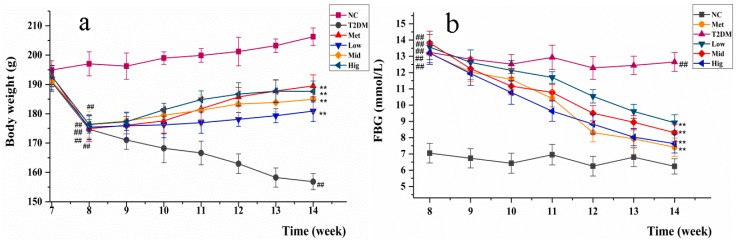
The changes in body weight (**a**) and fasting blood glucose (FBG) (**b**) of rats during the administration of Met and purified anthraquinone-Glycoside from *Rheum palmatum* L. (PAGR). NC, normal group; T2DM, type 2 diabetes mellitus; Met, metformin group; Low, low group (100 mg/kg); Mid, middle group (200 mg/kg); Hig, high group (400 mg/kg). The data were expressed as mean ± standard deviation (SD) (*n* = 10), ^#^
*p* < 0.05 ^##^
*p* < 0.01 vs.NC group; * *p* < 0.05 ** *p* < 0.01 vs. T2DM group.

**Figure 3 molecules-24-01454-f003:**
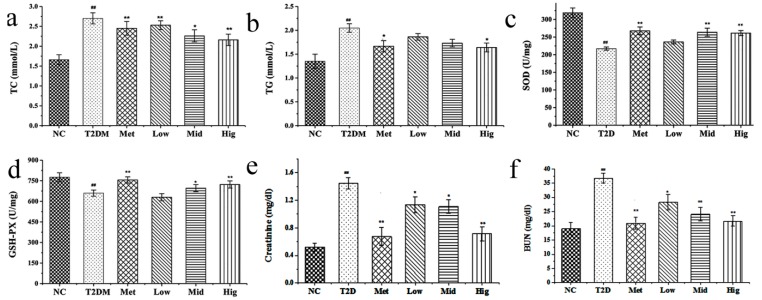
Effects of purified anthraquinone-Glycoside from *Rheum palmatum* L. (PAGR) on total cholesterol (TC) (**a**), triglyceride (TG) (**b**), superoxide dismutase (SOD) (**c**), glutathione peroxidase (GSH-PX) (**d**), creatinine (**e**), and BUN (**f**) in serum of rats. NC, normal group; T2DM, type 2 diabetes mellitus; Met, metformin group; Low, low group (100 mg/kg); Mid, middle group (200 mg/kg); Hig, high group (400 mg/kg). The data were expressed as mean ± SD (*n* = 10), ^#^
*p* < 0.05, ^##^
*p* < 0.01 vs. NC group. * *p* < 0.05, ** *p* < 0.01 vs. T2DM group.

**Figure 4 molecules-24-01454-f004:**
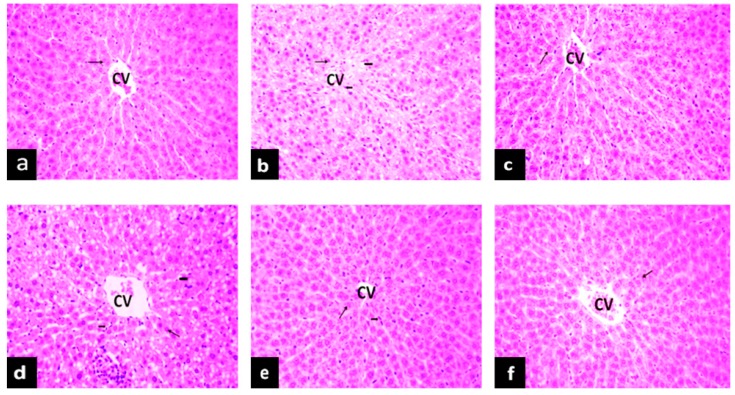
The effects of purified anthraquinone-Glycoside from *Rheum palmatum* L. (PAGR) on histopathological changes in the liver of rats. Histological observation, H&E, a–f×200; (**a**), normal (NC) group; (**b**), type 2 diabetes mellitus (T2DM) group; (**c**), metformin (Met) group; (**d**), Low group; (**e**), middle (Mid) group; (**f**), high (Hig) group; CV, central veins; 
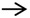
, hepatocyte; 
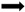
, inflammatory cell; 

, lipid droplet.

**Figure 5 molecules-24-01454-f005:**
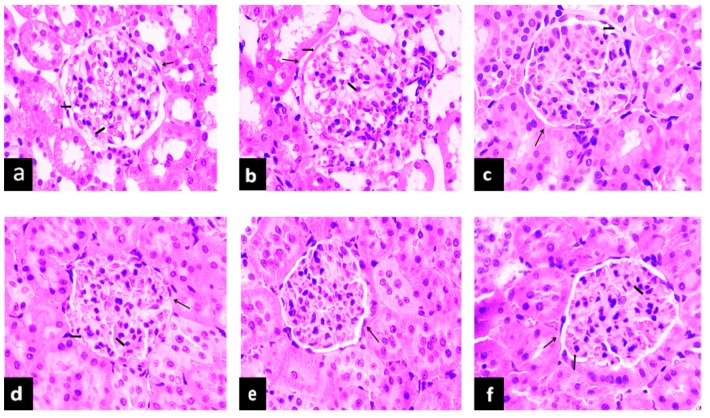
Effects of purified anthraquinone-Glycoside from *Rheum palmatum* L. (PAGR) on histopathological changes in the kidney of rats. Histological observation, H&E, a-f×400; (**a**), normal (NC) group; (**b**), type 2 diabetes mellitus (T2DM) group; (**c**), metformin (Met) group; (**d**), Low group; (**e**), middle (Mid) group; (**f**), high (Hig) group; 
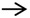
, glomerulus; 
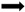
, base membrane; 

, mesangial.

**Figure 6 molecules-24-01454-f006:**
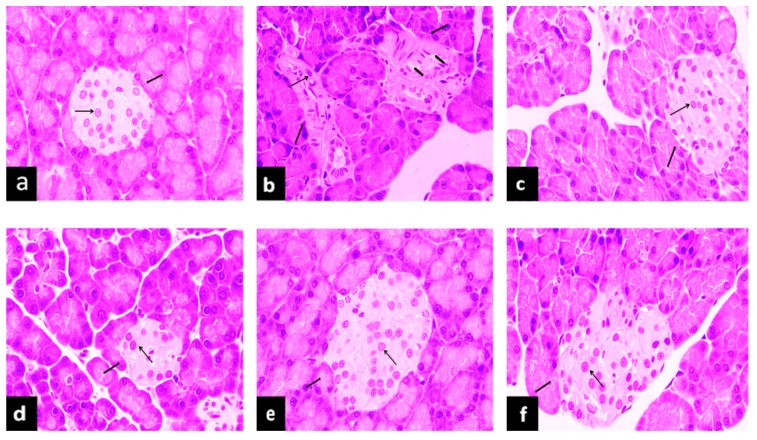
Effects of purified anthraquinone-Glycoside from *Rheum palmatum* L. (PAGR) on histopathological changes in the pancreas of rats. Histological observation, H&E, a–f×400; (**a**), normal (NC) group; (**b**), type 2 diabetes mellitus (T2DM) group; (**c**), metformin (Met) group; (**d**), Low group; (**e**), middle (Mid) group; (**f**), high (Hig) group; 
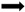
, islet; 
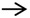
, islet cell; 

, lipid droplet.

**Figure 7 molecules-24-01454-f007:**
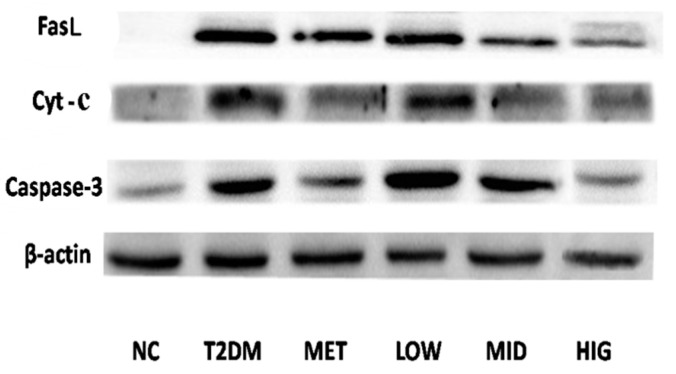
The effect of purified anthraquinone-Glycoside from *Rheum palmatum* L. (PAGR) on the protein expression of Fas ligand (FasL), cytochrome C (Cyt-c), and Caspase-3 in pancreatic tissue. NC, normal group; T2DM, type 2 diabetes mellitus; Met, metformin group; Low, low group (100 mg/kg); Mid, middle group (200 mg/kg); Hig, high group (400 mg/kg).

**Figure 8 molecules-24-01454-f008:**
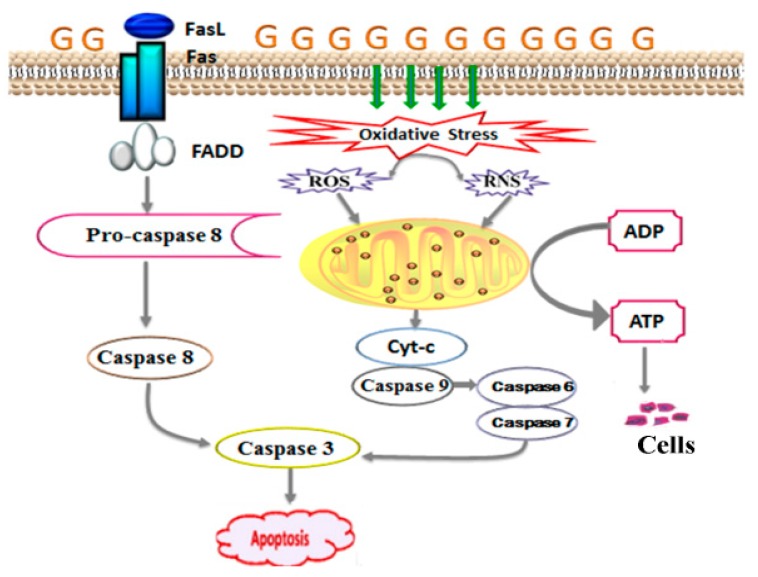
The signaling pathway of apoptosis. Hyperglycemia leads to the production of reactive oxygen species (ROS) and reactive nitrogen species (RNS) which change the membrane potential of mitochondria, leading to the release of cytochrome C (Cyt-c). It also leads to the expression of Fas ligand (FasL) and Fas, and they combine to form the Fas-associated death domain protein (FADD). The Cyt-c and FADD all activate caspase cascade reaction, leading to apoptosis.

## Data Availability

The data used to support the findings of this study are included in the article.

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
