# Peer review of "Ameliorative Effect and Mechanism of the Purified Anthraquinone-Glycoside Preparation from Rheum Palmatum L. on Type 2 Diabetes Mellitus"

_molecules, 2019, doi:10.3390/molecules24081454_

Round 1

Reviewer 1 Report

Dear authors,

Congratulations on the manuscript "Ameliorative effect and mechanism of the purified anthraquinone-glycoside preparation from Rheum palmatum L. on type 2 diabetes". However, I believe that some points can be revised to an improved final format.

In the figures the statements must be totally rewritten inserting complete legends for the treatments, as well as evaluated biological effects.

Figure 2 is low-set and needs to be adequate. I suggest changing the colors / fill of the bars. Also, the signals for statistical differences should be better defined.

There is no description of the botanical identification so little, from the herbarium deposit of the plant studied.

Also, no description of root particle size and filtration process not indicated.

At what temperature was the hydroalcoholic extract concentrated? Was rotavaporator used under reduced pressure?

What is the volume of extract that has been partitioned? What were the volumes of the solvents of increasing polarity?

What was the yield of the n-butanol fraction relative to the roots or to the hydroalcoholic extract? All of these details are important for the reproduction of searches!

How many subfractions were obtained from the fractionation?

The method for quantification of total anthraquinone-glycoside content should be indicated.

There is no project approval number on the Ethics Committee for Animal Use.

The antioxidant and biochemical analyzes need a brief description and bibliographic references should be added.

Author Response

Response to Reviewer

Reviewer 2 Report

The submitted paper describes ameliorative effect and mechanism of the purified anthraquinone – glycoside preparation from Rheum palmatum L. on type 2 diabetes. In my opinion, the study is valuable and the presented results are convincing.

There are only several minor points which should be addressed by the authors.

- Page 3, line 60: additional literature (if available) should be added to support this thesis

- Figure 2II: Y axis title should be improved

- Abbreviations should be defined at first mention and used consistently thereafter

- Page 9, line 179: ‘Figure 6 demonstrates that the islet shape (…)’ - this sentence should be improved

- Page 11, lines 205, 207 and 218: the symbols I and II referring to Figure 7 are too large; there should be a full stop after ‘Figure 7 II’

- Page 12: The caption of Figure 7 should be improved  

- Material and methods: the companies’ names, the names of the cities – capital letters should be used

- Material and methods: page 13, line 240: ‘chromatographically or analytically pure’

- Material and methods: page 14, line 266: ‘µ’ should be removed

- Material and methods: page 15, line 297: the title of the paragraph should be improved

- Material and methods: page 17, line 330: the first sentence should be improved

- References should be carefully checked and unified

Author Response

Response to Reviewer

Reviewer 3 Report

The manuscript by Cheng, et al. studied the potential anti-diabetic effects of PAGR in a type 2 diabetes rodent model. Here, the authors demonstrated that administration of PAGR decreased blood glucose levels and improved dyslipidemia. Besides, PAGR treatment also corrected pathological changes in the liver, kidney and islet. The authors also suggested that PAGR exerted the protective effects through enhancing anti-oxidant and anti-apoptosis capacity. Overall, this is an interesting pre-clinical work. However, before the current manuscript is considered to be accepted, there are some issues required to be addressed.

Major general issues:

1. One of the major undefined questions of this manuscript is that which organ is the primary targeted tissue for PAGR is unclear. From the results, the authors might consider islet β-cells would be mostly affected by PAGR treatment and result in amelioration in diabetic conditions. However, the authors need to provide more evidence to support their conclusion. Histological experiments such as insulin/glucagon staining as well as β-cell functional studies such as GSIS are recommended.

2. The other major issue is that the authors should also examine the changes in adipose tissues, given that adipose tissues are believed to play an important role in type 2 diabetes and metabolic defects. It is reasonable to look at the histology of white adipose tissues and detect inflammatory/fibrotic markers in adipose tissues.

Specific issues:

3. In Figure 3, SOD and GSH-PX contents in the serum have been demonstrated. It would be more informative if the authors could measure these parameters in different tissues, especially in the islet (β-cells).

4. In Figure 4:

1) I suggest using the same magnification for different groups;

2) The current arrows are very difficult to distinguish;

3) Could the authors show lipid accumulation in the liver?

5. In Figure 5:

1) From the histology, I couldn’t tell significant differences between groups. The authors need to provide clearer images to show more obvious histological changes;

2) In addition to the histological results, renal function parameters such as serum creatinine and BUN need to be shown.

6. For Figure 6:

1) I also suggest using the same magnification for different groups;

2) It is recommended to show insulin staining images to better prove the conclusion.

7. In Figure 7:

1) For Cyc-c, it is recommended to show its expression in mitochondrial and cytoplasmic fractions other than in the whole tissue lysate;

2) For caspase-3, it is recommended to detect active caspase-3 but not only total caspase-3.

8. The writing needs to be improved and polished.

Author Response

Response to Reviewer

Reviewer 4 Report

Manuscript Review Comments

Title: “Ameliorative effect and mechanism of the purified anthraquinone- glycoside preparation from Rheum palmatum L. on type 2 diabetes” (molecules-464626)

The authors present an animal study on the effects of an extract of the plant Rheum palmatum L. on type 2 diabetes mellitus. They state that treatment with PAGR ameliorates several diabetes and metabolic variables of rats. Although interesting, I find several shortcomings in the manuscript that should be addressed.

Comment to the authors

General comments

-          Article structure does not follow the structure of the template. Please adjust the manuscript structure following the manuscript template (e.g. results and discussion in separate sections).

-          The manuscript needs a grammar and style revision, maybe by a native English speaker. Several errors have found throughout the text.

-          Please check the acronym use of first mention on the manuscript.

Specific comments:

Title page and abstract:

-          Rheum palmatum L. should be specified in italics in the title.

-          Abstract structure could be improved by making a greater description of material & methods.

-          Please ensure that the keywords are correct MeSH terms. Use the appropriate terms as specified in MeSH for better indexation.

Introduction

-          Line 57: I do not find this expression to be correct on a scientific manuscript. Please correct.

-          If the authors are focusing their study on T2DM, they should avoid speaking of the type 1 diabetes since they are 2 different pathologies.

-          I think that introduction section needs a better rationale for the study.

-          In the text the term “type 2 diabetes mellitus” is used and hence should be also used in the title.

Results and Discussion

-          On what basis were the toxicity study concentration and the final concentrations used decided by the authors?

-          Figure 2 is not well understood. Needs a better description, names for the graphs etc.

-          Authors have put together both section and do not clearly discuss some results of the study. These two sections should go separately and each result should be discussed in a structured way.

-          Figure 4 needs a better format. Photographs are too small to be well interpreted. Figure legend is confusing and should be better explained. Same is applicable con figures 5 and 6.

-          No results of the study are shown in tables, which rises doubts about the validity of the data only with figure images and without the results of the statistical tests.

-          All the results based on histological images do not provide any quantification or specific markers. Maybe immunohistochemical images would better reflect some of the effects of the compound. Most part of the discussion is based on these images instead of the quantitative results of the study, making the article quite long.

Materials and Methods

-          Please comment on the study design and follow an appropriate guideline for the manuscript preparation and specify this in the methods section. In this case, ARRIVAL guidelines are the appropriate for animal studies.

-          Please give the correct descriptions for commercial references (e.g. Sigma Aldrich has not the correct reference).

-          3.6 Biochemical measurements: Please give the correct commercial descriptions of the kits.

-          Please provide a proper approval reference number of the ethical review board. Some authors from different affiliations and the animals and other procedures were purchased in other institutions. Were all the experiments performed in the institution that validated the study?

-          Statistical tests should be described in more detail: Was normality assessed? Since there are no tables, statistical tests cannot be defined in table footnotes.

References

-          Some references format needs to be checked for errors (e.g. references 9 and 33).

Author Response

Response to Reviewer

Round 2

Reviewer 1 Report

Dear authors,

With the adjustments made, I believe that the manuscript can be accepted for publication in the journal Molecules.

Congratulations.

Reviewer 3 Report

Although there are improvements compared to the original version, considering there are several major points the author failed to address, I would not recommend publication for the current manuscript.